# Assessment of Carbon Substrate Catabolism Pattern and Functional Metabolic Pathway for Microbiota of Limestone Caves

**DOI:** 10.3390/microorganisms9081789

**Published:** 2021-08-23

**Authors:** Suprokash Koner, Jung-Sheng Chen, Bing-Mu Hsu, Chao-Wen Tan, Cheng-Wei Fan, Tsung-Hsien Chen, Bashir Hussain, Viji Nagarajan

**Affiliations:** 1Department of Biomedical Sciences, National Chung Cheng University, Chiayi City 621, Taiwan; suprokashkoner@alum.ccu.edu.tw (S.K.); bashir.aku@gmail.com (B.H.); 2Department of Earth and Environmental Sciences, National Chung Cheng University, Chiayi City 621, Taiwan; cwfan@eq.ccu.edu.tw (C.-W.F.); mathumitha08@gmail.com (V.N.); 3Department of Medical Research, E-Da Hospital, Kaohsiung 824, Taiwan; nicky071214@gmail.com; 4Center for Innovative on Aging Society (CIRAS), National Chung Cheng University, Chiayi City 621, Taiwan; 5Division of Cardiology, Department of Internal Medicine, Ditmanson Medical Foundation Chiayi Christian Hospital, Chiayi City 600, Taiwan; kobebrnt2000@gmal.com; 6Department of Internal Medicine, Ditmanson Medical Foundation Chiayi Christian Hospital, Chiayi City 600, Taiwan; cych13794@gmail.com

**Keywords:** microbial community, Biolog EcoPlate™ assay, 16S RNA amplicon, hylogenetic investigation of communities by reconstruction of unobserved states (PICRUSt), functional metabolic pathway prediction, limestone

## Abstract

Carbon utilization of bacterial communities is a key factor of the biomineralization process in limestone-rich curst areas. An efficient carbon catabolism of the microbial community is associated with the availability of carbon sources in such an ecological niche. As cave environments promote oligotrophic (carbon source stress) situations, the present study investigated the variations of different carbon substrate utilization patterns of soil and rock microbial communities between outside and inside cave environments in limestone-rich crust topography by Biolog EcoPlate™ assay and categorized their taxonomical structure and predicted functional metabolic pathways based on 16S rRNA amplicon sequencing. Community level physiological profiling (CLPP) analysis by Biolog EcoPlate™ assay revealed that microbes from outside of the cave were metabolically active and had higher carbon source utilization rate than the microbial community inside the cave. 16S rRNA amplicon sequence analysis demonstrated, among eight predominant bacterial phylum Planctomycetes, Proteobacteria, Cyanobacteria, and Nitrospirae were predominantly associated with outside-cave samples, whereas Acidobacteria, Actinobacteria, Chloroflexi, and Gemmatimonadetes were associated with inside-cave samples. Functional prediction showed bacterial communities both inside and outside of the cave were functionally involved in the metabolism of carbohydrates, amino acids, lipids, xenobiotic compounds, energy metabolism, and environmental information processing. However, the amino acid and carbohydrate metabolic pathways were predominantly linked to the outside-cave samples, while xenobiotic compounds, lipids, other amino acids, and energy metabolism were associated with inside-cave samples. Overall, a positive correlation was observed between Biolog EcoPlate™ assay carbon utilization and the abundance of functional metabolic pathways in this study.

## 1. Introduction

Microbes-associated biomineralization is a widespread phenomenon in regions rich in limestone, and ultimately leads to the precipitation of calcium carbonate [1]. In this process, the available nucleation sites are one of the prime governing factors along with the concentration of calcium ions, dissolved inorganic matter, and the pH of that particular atmosphere [2]. As such, the growth and metabolic activity of microbes can ubiquitously speed up the biomineralization process by providing more nucleation sites for CaCO_3_ precipitation in such an environment [3,4]. The efficiency of exogenous carbon source utilization from soil organic carbon (SOC) could be an embracive signature of microbial community metabolic characteristics and their growth [5]. Thus, previous studies reported that the growth of bacteria cells and their metabolism are strongly influenced by diverse carbon source catabolism activity, because it may deliver the building block metabolites and energy for them [6,7]. In this aspect, the pool of SOC might be different according to the variations of natural earth crust structures, which are promotionally allied with the amount of soil microbial biomass [8,9]. However, the assimilation or uptake of carbon from the environment is considered the key factor for microbial-mediated calcification in limestone [10].

In terrestrial limestone-rich habitats, the majority of heterotopic calcite-precipitated bacteria play a role in the carbon cycle via the utilization of soil organic matter (SOM) from organic matter [11]. Based on its molecular mass, SOM can be traditionally categorized as either low-molecular-weight (LMW) or high-molecular-weight (HMW) [12]. Furthermore, carbohydrates, organic/amino acids, proteins, siderophores, lipids, phenolics, hormones, and vitamins, among others, can be categorized into different classes of LMW compounds, and have relatively less persistence in terrestrial ecosystems due to their higher uptake and metabolisms by soil microbiota [13]. These are formed during the partial decomposition and conversion of plant inputs, such as root exudates and upper and lower-ground litter by soil organisms [14]. Caves are considered extreme environments due to the absence of sunlight. Here, no photosynthesis occurs, resulting in oligotrophic conditions [15]. Non-photosynthetic primary production is always carried out by several chemoautotrophic microbes and supporting other bacteria groups to be sustained [16]. The previous report regarding such an environment emphasized that some heterotopic bacteria, which might be in the presence of Ca ions and CO_2_, can construct calcium carbonate via decomposition of urea by urease enzyme [17,18]. In such nutrient-limited ecosystems, a small amount of allochthonous carbon can accumulate with the help of surface runoff and vadose-zone leaching from photic surface environments, including macrofaunal activity, which act as pioneer organic matter sources, leading to the microbial biomineralization of calcium carbonate [19,20].

As substrates of Biolog EcoPlate™ also belong to LMW carbon compounds, it would be a promising technique and powerful analytical tool to determine the carbon catabolism ability outlook of heterotopic bacteria during their cell growth process in plate conditions [21]. This approach characterizes the metabolic diversity of environmental samples using community-level physiological profiling (CLPP) [22,23]. The Biolog EcoPlate™ assay plate consists of 31 most-useable carbon sources in triplicate order with tetrazolium redox violet dye; during each substrate utilization through the inoculated microbiota community, the dye converts into a purple color and denotes as subsequent carbon soles are utilized [24]. Next-generation sequencing (NGS) is a revolutionary breakthrough, and uses present decays as a robust culture-independent technique to explore the biodiversity of complex environmental niches based on 16S rRNA targeting specific regions of DNA sequences [25]. This 16S rRNA gene amplicon sequence is used to obtain prediction-based phylogenetic structures of the uncultivated microbial community, metabolic pathway, functional genes, and species diversity studies [26]. The phylogenetic investigation of communities by the reconstruction of unobserved states (PICRUSt) analysis is a computational approach to predict the functional potentiality of microbial groups based on 16S rRNA hypervariable amplicon sequencing data. A reference genomic database was generally linked with PICRUSt analysis to predict the functional role of unculturable prokaryotes in a precise ecosystem [27,28]. Together with Biolog EcoPlate™ substrate utilization, 16S rRNA amplicon-based functional prediction data would be a compact approach to explore the overall view on the microbial community’s metabolic fingerprint.

Organic matter has been reported to influence the microbial community associated with biomineralization by several bacterial species [1]. Still, there is a lacuna on how it can affect upon a microbiome’s communal metabolic fingerprint in such an environment. Currently, no studies have characterized the microbiota inside and outside of caves in a limestone-rich region using a community-level physiological profiling (CLPP) approach in combination with 16S rRNA amplicon-based functional prediction analysis. In the present study, the carbon catabolism patterns of microbial communities inside and outside a limestone cave, as well as their functional diversity from a holistic point of view, were investigated. To this end, the Biolog EcoPlate™ assay was used to determine the carbon assimilation ability of limestone microbiota. Additionally, NGS was applied to the taxonomic structure of the microbial community, and PICRUSt2 analysis was performed using the KEGG (Kyoto Encyclopedia of Genes and Genomes) reference database to predict the metabolic pathways of the microbial communities studied.

## 2. Materials and Methods

### 2.1. Sample Collection Information

The sampling site was located in the Tianliao district of Southern Taiwan. Rock and soil samples were obtained from two different zones of the limestone-rich region, according our sampling criteria: outside a limestone cave (22°50′40″ N; 120°23′10″ E) and inside a limestone cave (22°50′40″ N; 120°23′10″ E), shown in Figure 1. In addition, specific meteorological parameters, such as solar radiation intensity, ambient temperature, humidity, and soil temperature, from the region of sample collection were recorded. Details of the sampling information are provided in Appendix A. The rock samples were collected from the surface area of the mother rock and stored in a pre-sterilized zipper bag to avoid contamination. The soil samples were collected from beneath the topsoil (O horizon) zone corresponding to the rock samples collected from the surrounding area and stored as above. Ultimately, all samples were transported directly to the laboratory using a temperature-controlled box.

### 2.2. Carbon Substrate Utilization Pattern

Community-level carbon substrate utilization patterns were determined for all soil and rock samples using a commercial phenotyping microarray tool (Biolog EcoPlate™ Inc., Hayward, CA, USA). For sample pre treatment, an improved version of the experimental protocol was used, according to the manufacturer’s instructions. Briefly, 2 g of soil and rock samples were taken separately into 30-mL sterilized conical tubes, followed by 10 min of vortexing with 19 mL of 1X sterilized phosphate buffer solution (PBS). The sample tubes were mixed homogeneously with PBS by shaking for up to 1 h to release bacterial cells, followed by 10 min of holding under static conditions to allow the soil particles to settle down. Then, 2 mL of supernatant was taken carefully and a serial dilution of up to 10-3-fold was performed to reduce the microbial load to obtain an efficient result and consider similar cell concentration in each sample. Finally, 100 µL of suspension from the 10-3-fold dilution tube was inoculated into each well of a Biolog microplate and incubated under dark conditions at 25 °C for 168 h. A suspension of pure cultured *E. coli* with the same dilution as the sample was used as a positive control, while double-distilled water was used as a negative control, and inoculated into the corresponding microplate wells.

To determine the substrate utilization pattern of the microbial communities sampled, the optical density (OD) of each incubated microplate was observed every 24 h at a wavelength of 590 nm using a microplate reader (Multiskan™ FC Microplate Photometer; Thermo Scientific, Loughborough, UK) [22]. Mathematical equations were used to calculate the average well color development (AWCD) for each microplate, according to a previous study [29]. The OD readings against each well were corrected using the OD reading of water as a control. After correction, the negative readings were adjusted to zero. Lastly, based on substrate utilization patterns by microbes, the substrate utilization Shannon diversity substrate richness (H) and evenness (D) were calculated, according to previous studies [21,24,30].

### 2.3. Microbial Genomic DNA Extraction

In order to obtain homogenized microbial genomic DNA, each sample was evenly mixed, and large particles were segregated using a sieve to ensure complete homogenization and DNA extraction. Then, 0.5 g of fine powder from the rock and soil samples were inoculated into respective lysis tubes for the extraction of gDNA as per the protocol of NucleoSpin^®^ Soil, a commercial kit for gDNA extraction (MACHEREY-NAGEL GmbH & Co., Düren, Germany), with some modifications: instead of single step elution, two elution times were performed using 50 µL of SE buffer, followed by 1 min incubation and 30 s of vortexing to obtain the maximum volume of DNA. Finally, the quantification and quality of gDNA was assessed at 260/280 wavelength using a Nanodrop 1000 (Thermo Fisher Scientific, Waltham, MA, USA). The extracted gDNA was stored at −20 °C until further NGS analysis.

### 2.4. 16S rRNA Amplicon Library Preparation and Functional Prediction

The amplification of the 16S rRNA gene targeting V3-V4 regions gene sequence was carried out by using the KAPA HiFi HotStart Ready Mix PCR kit, and primer information is shown in Appendix A. Amplicon PCR amplification was performed using Illumina’s MiSeq system (Illumina, San Diego, CA, USA) following a comprehensive protocol for paired-end sequencing, as described in a previous study [31]. After obtaining the raw sequencing files from the NGS platform, the data were further analyzed using the Qiime2 software package to characterize the microbial diversity at the phylum and genus levels [32]. In short, the sequence data were trimmed to remove chimera and clustered into Amplicon Sequence Variant (ASV) using a 97% threshold limit of similarity with respect of Greengenes database. Lastly, the Qiime2 view was used to determine the bacterial composition of each sample. Additionally, PICRUSt2 software based on the Kyoto Encyclopedia of Genes and Genomes (KEGG) was used to predict the functional pathways associated with the microbial taxa within each community, as described previously (http://picrust.github.io/picrust/, accessed on 1 May 2020) [33].

### 2.5. Data Visualization and Statistical Analysis

The Student T-test was used to show a statistically significant difference between outside and inside-cave microbiota substrate utilization AWCD value, Shannon index (H’), Simpson index (D), and Shannon evenness score in soil and rock samples. Additionally, Pearson’s correlation analysis was performed using SPSS (https://www.ibm.com/products/spss-statistics, accessed on 5 May 2020) to determine the relationship between carbon source utilization and abundance of predicted functional pathway reads of the microbial communities. To determine the association between bacterial taxonomies and functional predictions, heatmaps with dendrograms were computed using the web-based tool clustvist (https://biit.cs.ut.ee/clustvis/, accessed on 10 May 2020) and a morpheus heatmap (https://software.broadinstitute.org/morpheus/, accessed on 13 May 2020). Principal component analysis (PCA) was conducted using canoco5.1 software (http://www.canoco5.com/, accessed on 14 May 2020). The variations in the functional metabolic pathways between each sample after PICRUSt2 analysis were visualized using excel 2007 accessed on 14 May 2020.

## 3. Results

### 3.1. Variation in Average Well Color Development (AWCD) during Incubation Period

The average well color development (AWCD) curve was plotted during the incubation period after the inoculation of the samples into the Biolog EcoPlate™, as shown in Figure 2. The microbial communities of the samples collected from outside the cave (LR-01 and LS-01) exhibited the highest AWCD kinetics, suggesting that the microbiota of these samples were highly active in their use of different types of carbon sources during cell growth. Besides a higher substrate utilization, Shannon diversity index (H) were found in this area’s samples at the end of the incubation period (168 h), shown in Table 1. The microbial community in this area also indicated that they have a shorter lag phase period of cell growth. In addition, substrate utilization Shannon evenness scores in these samples were 0.86 and 0.97 (Table 1). Conversely, rock and soil samples from inside the cave showed lower AWCD rates than those collected outside of the cave (*p* < 0.05). This suggests that the microbial community in this sampling zone was lacked the activity to utilize different types of external carbon sources during cellular growth, resulting in a longer lag phase, as observed in both samples. In particular, the rock microbiota sample required around 96 h to consume a significant amount of Biolog EcoPlate™ substrates compared with the soil sample, which was markedly faster (24–48 h). Furthermore, substrate utilization Shannon diversity index and evenness scores of both the rock and soil samples in this area were low (*H* = 2.17, 2.65; Shannon evenness = 0.62, 0.76; *p* < 0.05). This analogous trend of carbon substrate utilization between different sampling zones suggests that microclimate conditions have an effect on the carbon source assimilation pattern of microbial communities. Additionally, the positive control showed a certain AWCD value during the incubation period, while no AWCD value was found for the negative control.

### 3.2. Major Carbon Source Assimilation Pattern at End Point (168 h)

The 31 types of carbon substrates found in Biolog EcoPlate™ were grouped into six major categories according to their biochemical properties: carbohydrates, amino acids, carboxylic acids, polymers, amines, and phenols. The utilization capability of these six categories, which was highest in the samples from outside the cave compared with the samples taken from inside the cave, where the utilization rate was quite low, is shown in Figure 3. The microbial communities in the rock and soil samples collected from outside the cave were found to utilize carbohydrates, amino acids, carboxylic acid, and polymers more commonly than amines and phenols. However, the higher degree of carbohydrate and carboxylic acid utilization was more closely associated with the soil microbiota than the rock microbiota. By contrast, although the microbial communities of the rock and soil samples collected from inside the cave initially consumed less of the major groups of carbon substrates, they showed a considerable utilization at the end of the incubation period (168 h), indicating that these microbes are also able to catabolize carbohydrates, amino acids, carboxylic acids, and polymers with a significantly slower utilization rate during cellular growth. Furthermore, the microbial communities of the soil samples showed a faster rate of amino acid, carbohydrate, and carboxylic acid assimilation than the rock samples inside the cave, followed by amines and phenols. Notably, polymers were used up more by the microbiota of the rock samples than the microbiota of the soil samples inside the cave.

### 3.3. Analysis of Bacterial Community Compositions and Distribution Patterns Using 16S rRNA Amplicon-Based Techniques

The bacterial compositions of the microbiota of the soil and rock samples both outside and inside the cave are shown in Figure 4 at the phylum level. A total of 27 classified taxa at phylum level were detected using 16S rRNA amplicon data across all samples, with similar beta diversity patterns (Appendix A). Among these, Proteobacteria, Acidobacteria, Actinobacteria, and Planctomycetes were the most predominant, followed by Nitrospirae, Chloroflexi, and Gemmatimonadetes in all of the samples, except Cyanobacteria, which was only present in the samples collected outside the cave. However, the relative abundance of Cyanobacteria was higher in the soil samples (9.33%) than the rock samples (2.59%). The relative abundance of Proteobacteria, Planctomycetes, Actinobacteria, and Acidobacteria was 12.75% and 13.60%, 26.61%, and 32.07%, 18.23% and 6.94%, and 8.08% and 13.05% in the samples collected from rock and soil outside the cave, respectively. By comparison, the relative abundance of Actinobacteria, Acidobacteria, and Proteobacteria were 26.98% and 21.37%, 17.68% and 17.81%, and 21.97% and 21.67% in the rock and soil samples collected inside the cave, respectively. Pearson’s correlation test was used to determine the effect of key environmental factors (e.g., sunlight, humidity, and temperature) on the distribution pattern of the bacteria identified. Planctomycetes, Proteobacteria, and Cyanobacteria were found to be significantly positively correlated with sunlight, while humidity was found to be negatively correlated (Appendix A). Actinobacteria showed an inverse relationship with sunlight and humidity compared with the abovementioned phyla. Dendrogram plot was constructed using a heatmap to visualize the distribution pattern of the bacteria at the genus level (Appendix A). As a result, two separates of clusters were observed, each denoting one of the two different zones of our sampling criteria: cluster 1 was found to be closely related to the bacterial genus diversity of the rock and soil samples collected from the inside cave, and cluster 2 represented the genus level distribution of the microbiota of the rock and soil samples collected from outside the cave. Thus, the different types of sampling zones showed similar types of bacterial genus diversity distribution patterns.

### 3.4. Correlation between Predicted Functional Pathways and Carbon Utilization Pattern

PICRUSt2 was used to predict the potential metabolic functions of the microbial communities in the samples using 16S rRNA amplicon sequence data. The aim of this analysis was to identify the KEGG pathways associated with the metabolism of LMW organic compounds indirectly involved in calcium carbonate formation. The functional prediction analysis revealed the presence of metabolic pathways, including carbohydrates, amino acids, other amino acids, and lipids, or xenobiotic compounds, including energy utilization and membrane transport, both within the samples collected from inside and outside the cave. Among these metabolic pathways, carbohydrate metabolism was found to be the predominant function, followed by amino acid metabolism, in all of the samples, as shown in Figure 5A. However, other metabolic functions, including membrane transport, energy metabolism, xenobiotics biodegradation, metabolism, and lipid metabolism were found in low proportions. Further categorization of the carbohydrate metabolism at KEGG level 3 indicated the presence of major metabolic functions related to starch, sucrose, galactose, amino sugars, nucleotide sugars, ascorbate, aldarate, the citrate cycle (TCA), and the pentose phosphate pathway in samples collected both inside and outside the cave as shown in Appendix A. Similarly, amino acid and other amino acid metabolisms were further categorized up to KEGG level 3, which included arginine, proline, phenylalanine, glycine, serine threonine, and glutathione as major functions. Lipid catabolism may be carried out by glycerophospholipid metabolism, while xenobiotic-related metabolism was utilized through bisphenol and benzoate degradation pathways. In addition, the carbon fixation pathway and methane metabolism were the main energy utilization pathways in the diversity structure of the microbiota from inside and outside the cave. Moreover, the ABC transporter was predominately found as a major function in the membrane transport pathway, which had a relatively higher abundance in the samples collected outside the cave compared to those collected inside the cave (Figure 5B). Furthermore, Pearson’s correlation statistical approach was used to evaluate the relationship between extracellular carbon source uptake and the utilization capabilities of the microbial community with their predicted metabolic functions, shown in Figure 6. The correlation analysis revealed that, among all predicted metabolic pathways, starch, sucrose, amino sugars, nucleotide sugars, glycerophospholipid metabolism, and benzoate degradation had a strong positive correlation with extracellular carbon utilization (*p* ≤ 0.05). Additionally, the carbon fixation pathway, methane metabolism, and ABC transporter were also positively correlated with extracellular carbon metabolic potential. However, several metabolic functions, including photosynthesis antenna proteins, phenylalanine, arginine, proline ascorbate, the aldarate pentose phosphate pathway, and galactose phosphate, were found to be negatively correlated with Biolog EcoPlate™ carbon substrate utilization.

### 3.5. Visualization of Multivariate Analysis of Eco Plate and 16S rRNA Amplicon Data

PCA analysis was conducted for the bacterial communities found both outside and inside the cave, performed with their major carbon source utilization pattern in Biolog EcoPlate™ against the incubation period (168 h) and predicted KEGG metabolic functions associated with carbon metabolism. Planctomycetes, Proteobacteria, Cyanobacteria, and Nitrospirae were strongly associated with the samples collected outside the cave, in which sunlight was a key determining factor (Figure 7). Additionally, the bacteria of this sampling zone were found to be involved in the utilization of the majority of carbon sources. These bacteria were predominantly associated with two important metabolic functions, namely the amino acid and carbohydrate metabolism. However, Acidobacteria, Actinobacteria, Chloroflexi, and Gemmatimonadetes were more closely associated with the soil and rock samples collected inside the cave. In Figure 7, the arrows, which represent the different types of carbon source utilization patterns, were not correlated with sunlight, but rather other environmental parameters, including humidity and temperature. The lipid metabolism, other amino acids, and energy metabolism pathways (e.g., carbon fixation pathways in prokaryotes and methane metabolisms) were found to be closely related with the evolution of the microbial communities and environmental information processing inside the cave.

## 4. Discussion

The use of Biolog EcoPlate™ is highly convenient to measure community-level microbial catabolic activity and diversity in terms of substrate utilization in a variety of complex environmental samples. However, it has some limitations, such as its ineffectiveness on aquatic samples, the sensitivity of redox dye against temperature, and its efficiency on heterotrophic bacteria [34]. However, in our field of study, it is rarely used and represents a completely novel approach [24]. In the present study, the microbiota from soil and rock samples collected outside of a limestone cave showed the highest rate of AWCD, compared with samples collected inside the cave, which showed a relatively low AWCD curve. In addition, fluctuations in the AWCD curves were also observed in the microbial communities of different sample types (rock and soil) within the same sampling zone. With regards to the difference observed in the AWCD patterns between sampling environments (i.e., outside and inside the cave), the microbial communities found outside of the cave were highly metabolically active, consuming and up taking carbon from the environment to fuel catabolic activity during cell growth. Previous studies have also reported that, in terrestrial environments, sunlight is the most influential factor for the production of simple organic carbon via photodegradation, resulting in a higher metabolic activity in microorganisms [35,36]. Similarly, in a study on permafrost, another extreme environment, the authors found that sunlight exposure can enhance the microbial respiration rate up to >40% compared with dark areas, mediated by dissolved organic carbon [37]. In this context, the absence of solar radiation may be responsible for the low levels of photodegradation within cave environments. This leads to the formation of oligotrophic conditions, which ultimately inhibit organic carbon-mediated microbial functions and the catabolic rate of the microbial communities [37,38]. Thus, according to our results, the microbial communities inside caves poorly uptake carbon from external sources, resulting in lower AWCD curves. Additionally, due to their inefficient catabolic activity, these microbial communities may not be able to produce sufficient NADH during respiration, resulting in a reduced tetrazolium dye reduction in the Biolog EcoPlate™ plate during the 7-day incubation period, and a lower AWCD pattern [39].

In addition to the main six groups of carbon sources utilized by the microbial communities, our results showed that, among 31 carbon substrates, the utilization of carbon from carbohydrates, amino acids, and carboxylic acid groups was markedly higher in the microbial communities outside the cave. Although the microbial communities inside the cave also utilized these carbon substrates, the level of utilization was much lower than that of bacteria outside the cave. These results are in accordance with a previous study on soil microbial ecology, which found that carbohydrates, amino acids, and carboxylic acids belong to the LMW organic substance (LMWOS) class, which are preferentially taken up by bacteria found in soil for cellular activity, producing CO_2_ as a by-product [40]. These three groups of LMWOS are utilized by bacteria because of their high demand for carbon for anabolic products during cell mass development, which are also incorporated into the citric acid cycle, glycolysis, and the pentose phosphate pathway [29]. However, the dark conditions inside caves hamper sunlight-based primary productivity and the decomposition of these three major classes of LMWOS, leading to low concentrations inside caves, despite the input of allochthonous carbon from outside caves [41]. We suggest that the oligotrophic condition could contribute to the LMWOS stress situation in the inside-cave environment. Therefore, the microbial community of inside cave in this study was not adapted to the utilization of carbohydrates, amino acids, or carboxylic acid groups as carbon sources, and showed a low utilization pattern for these three groups of substrates.

According to Illumina-based 16S rRNA amplicon data analysis, Proteobacteria, Acidobacteria, Actinobacteria, and Planctomycetes were the predominant groups of phyla, followed by Nitrospirae, Chloroflexi, and Gemmatimonadetes, across all samples, irrespective of the sampling area. However, Cyanobacteria was the only phylum that was present in both rock and soil samples outside of the cave. Among these major actinobacteria, Acidobacteria, Chloroflexi, and Gemmatimonadetes were mainly associated with sampling inside the cave. This is in accordance with the results of a shotgun metagenomic study on the diversity of the microbial community inside a Manao-Pee cave in Thailand. This study found that actinobacteria and proteobacteria were predominantly associated with the limestone sample. However, in their study, Bacteroidetes, Firmicutes, Acidobacteria, Planctomycetes, Chloroflexi, Gemmatimonadetes, and Cyanobacteria were less abundant [15]. Additionally, another previous study based on metagenomic analysis of the microbial community of limestone reported a phylum level distribution and abundance similar to that found in the present study [42]. In addition, Meier et al., (2017) also found that a similar distribution pattern of microbial diversity at the phylum level associated with the microbial communities of soil and rock samples in regions rich in limestone. They found that Proteobacteria was predominant in the soil samples, while Proteobacteria, Actinobacteria, Bacteroidetes, and Firmicutes were predominant in the rock samples [43]. In the present study, among the identified phyla, Actinobacteria, Acidobacteria, Chloroflexi, and Gemmatimonadetes were found to be negatively correlated with sunlight, suggesting that these microbes use light-independent metabolic pathways. Microorganisms associated with dark and nutrient-poor cave areas are able to survive easily in this type of environment due to their chemoautotrophic mechanisms [44]. In fact, the majority of bacterial communities found in extreme environments are associated with nitrogen-, sulfur-, and methane-based chemoautotrophic pathways [45].

According to functional prediction analysis, the abundance of LMW organic compounds related metabolic pathways reads such as carbohydrates, amino acids, other amino acids, and lipids were predominating across all samples irrespective of the sampling area. This result suggests that microbial communities are able to utilize organic carbon sources for their metabolism. Previous studies have also demonstrated that the carbohydrate and amino acid metabolisms are the most dominant metabolic pathways associated with the microbial communities found in caves [15,27]. The higher contribution of these LMW organic compounds to the metabolic pathways of soil and rock samples collected outside the cave demonstrates that these were more functional than those from inside the cave, which is reflected by the utilization of the carbon sources from the Biolog EcoPlate™ by the different bacterial communities. Conversely, even though the bacterial communities inside the cave also had LMW organic compounds-related metabolic pathways, they did not show high levels of utilization activity for the different carbon sources in the Biolog EcoPlate™ assays. This suggests that oligotrophic situations inside caves ecosystems might be LMW organic compound-related metabolic pathways that are non-functional in real cases. Previous studies have found that the ABC transporter, which plays a key role in the uptake of LMW organic compounds from the environment, was relatively less abundant among the microbial communities found inside caves, which may explain their lower AWCD curves [46,47]. In addition, the methane-linked energy metabolic pathway, including a small amount of carbon fixation pathways, was found to be predominant in the bacterial communities of the cave samples. This could imply that bacteria found in caves can use methane as an alternative source of carbon, engaging in primary production via the chemolithoautotrophic pathway [15]. Previous studies have also shown that ABC transporter-associated genes are expressed at comparatively lower levels in autotrophic bacteria [48,49]. In this context, this study also considered the chemolithoautotrophic pathway, which is an autotropic pathway, dominant in the bacterial communities of inside-cave samples.

## 5. Conclusions

This study demonstrated that environmental factors greatly influence the taxonomic and physiological distribution of microbial communities in extreme ecosystems, such as limestone caves. These communities of bacteria use alternative sources of micronutrients from their environment for cellular growth, and ultimately participate in primary production to support higher-order organisms. The results of the Biolog EcoPlate™ assays showed that the microbial communities outside caves consumed simpler forms of different carbon sources, such as carbohydrates, amino acids, and carboxylic acids, than those inside caves. In contrast, specific bacterial groups, such as Actinobacteria, Proteobacteria, and Acidobacteria, could adapt to oligotrophic conditions by using alternative energy metabolic pathways, collectively known as chemoautotrophs, and were found to be the dominant phyla inside the caves. In the present study, the methane-based chemoautotrophic pathway was found to play a leading role in primary production by these bacterial communities, while the LMW organic compound-dependent primary production was probably non-functional in the dark oligotrophic conditions of inside-cave areas.

## Figures and Tables

**Figure 1 microorganisms-09-01789-f001:**
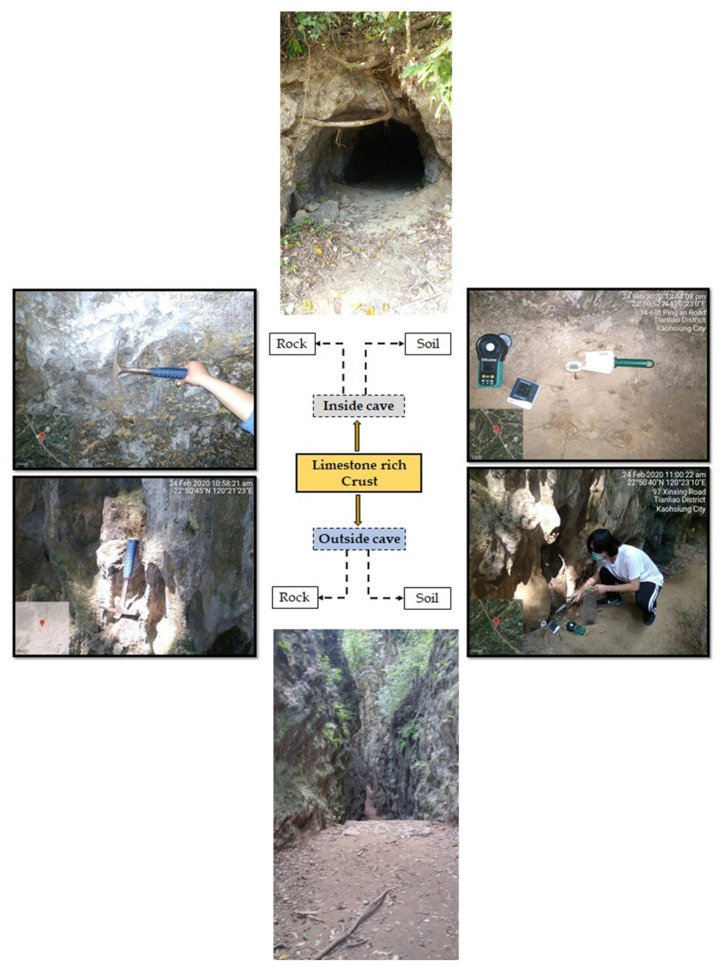
Sampling area and site description.

**Figure 2 microorganisms-09-01789-f002:**
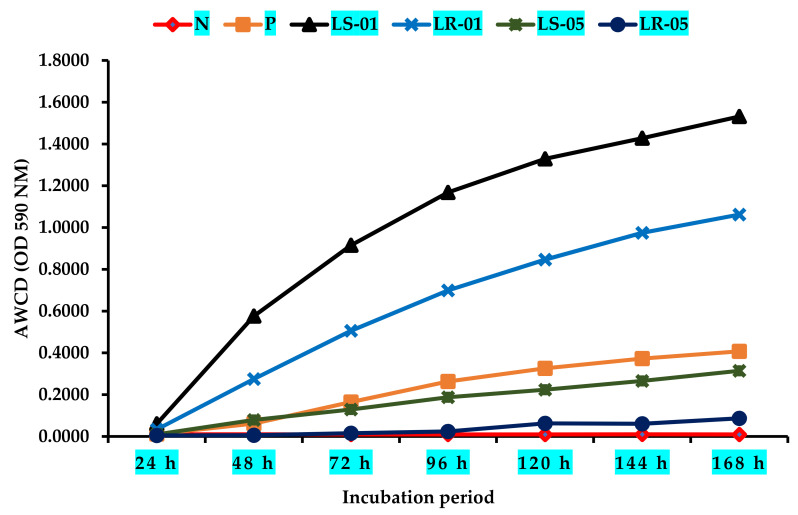
Changes in AWCD cure of all rock and soil samples according to different sample zones (LS1 and LR1, outside cave; LS5 and LR5, inside cave). N = negative control, P = positive control.

**Figure 3 microorganisms-09-01789-f003:**
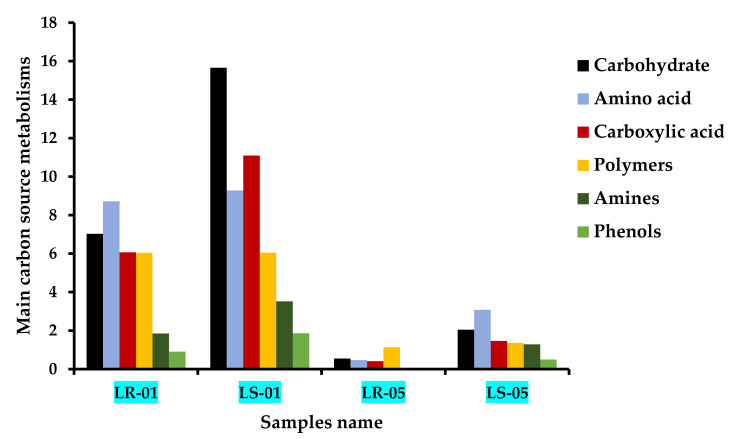
Major carbon source metabolic patterns at the incubation end point (168 h) of all rock and soil samples according to the different sampling zones (LS1 and LR1, outside cave; LS5 and LR5, inside cave).

**Figure 4 microorganisms-09-01789-f004:**
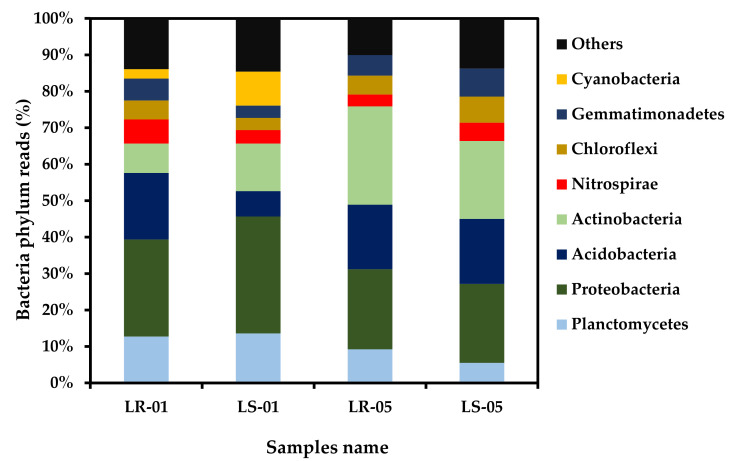
Major bacterial phylum relative abundance of rock and soil samples collected outside the cave (LR-01 and LS-01) and inside the cave (LR-05 and LS-05).

**Figure 5 microorganisms-09-01789-f005:**
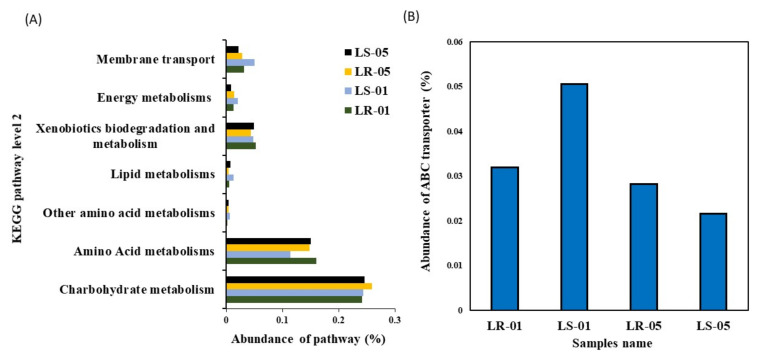
(**A**) Visualization of predicted major metabolic pathway categories relative to the abundance of each sample, including environmental information processing. (**B**) Presence of environmental information processing (ABC transporter) genes in samples collected outside and inside the cave using PICRUSt2 analysis.

**Figure 6 microorganisms-09-01789-f006:**
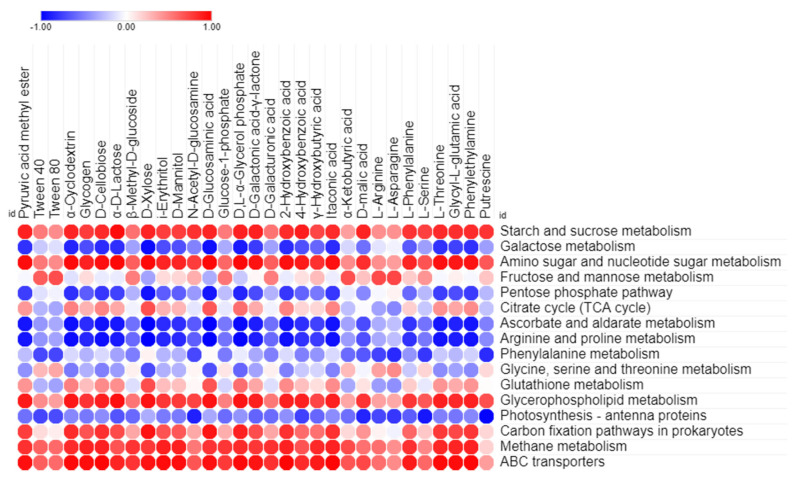
Visualization of Pearson’s correlation statistical test between 31 Biolog EcoPlate™ carbon sources utilization and related predicted functional KEGG pathway abundance of bacterial communities in all over the samples.

**Figure 7 microorganisms-09-01789-f007:**
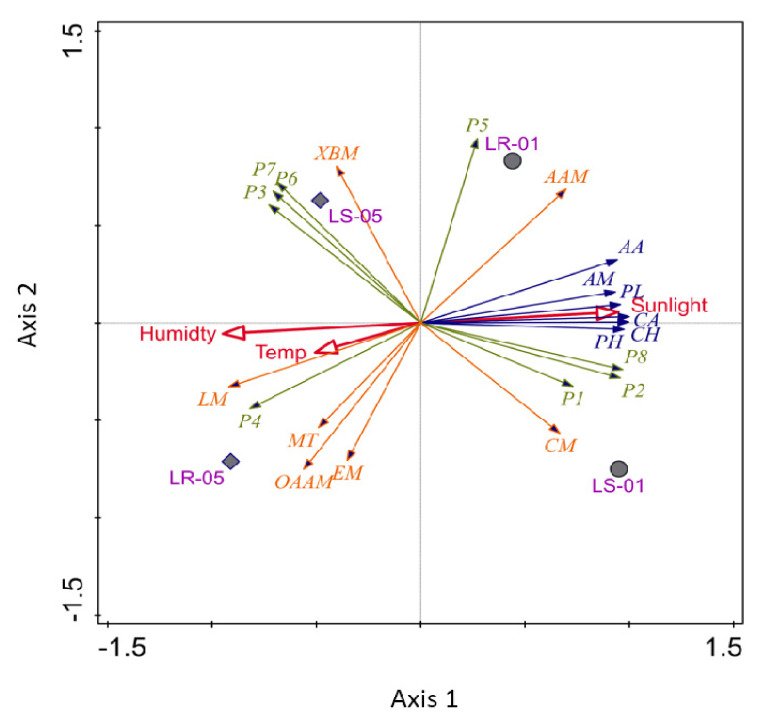
Principle component analysis of major carbon sources (AA, amino acids; AM, amines; PL, polymers; CA, carboxylic acid; CH, carbohydrates; PH, phenols) with predicted KEGG pathways (XBM, Xenobiotic biodegradation and metabolisms; AAM, amino acid metabolism; CM, carbohydrate metabolism; LM, lipid metabolism; EM, energy metabolism; OAAM, other amino acid metabolism; MT, membrane transport), including major bacterial phylum (P1, Planctomycetes; P2, Proteobacteria; P3, Acidobacteria; P4, Actinobacteria; P5, Nitrospirae; P6, Chloroflexi, P7, Gemmatimonadetes; P8, Cyanobacteria) against the environmental parameters.

**Table 1 microorganisms-09-01789-t001:** Substrate utilization diversity and evenness index among rock and soil samples according to the different sampling zones.

Parameters	LR-01	LS-01	LR-05	LS-05	*p* Value
AWCD	1.06	1.53	0.09	0.31	<0.05
Shannon Index H’	3.00	3.38	2.18	2.65	<0.05
Simpson Index D	0.98	0.98	1.00	0.98	>0.05
Shannon evenness	0.87	0.98	0.63	0.77	<0.05

LR-01, LS-01: outside the cave; LR-05, LS-05: inside the cave.

## Data Availability

The data presented in this study are available on request from the corresponding author.

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
