# Peer review of "Assessment of Carbon Substrate Catabolism Pattern and Functional Metabolic Pathway for Microbiota of Limestone Caves"

_microorganisms, 2021, doi:10.3390/microorganisms9081789_

Round 1
Reviewer 1 Report
Review and comments to the manuscript ID microorganisms-1334432 and titled “Assessment of carbon substrate catabolism pattern and functional metabolic pathway for microbiota of limestone caves”
Authors: Suprokash Koner et al.
In fact, underground ecosystems are poorly understood in terms of microbiology. Therefore, I always support the study of the cave microbiota and mycobiota, especially if the research goes beyond the usual assessment of species composition. In my opinion, this manuscript is very interesting and well written. I have no typical substantive comments and congratulations on the author of the idea and the manuscript. Nevertheless, I ask the authors to read the manuscript carefully and correct any minor flaws before accepting the manuscript for publication. I have listed some of them below:
1) Ln 49: „...sites for CaCO3 precip-...”
“3” - subscript
2) Figure 1: Please remove "hrs" from the chart axis.
3) Figs. 4 and 5: not readable
Author Response
Review and comments to the manuscript ID microorganisms-1334432 and titled “Assessment of carbon substrate catabolism pattern and functional metabolic pathway for microbiota of limestone caves”
Authors: Suprokash Koner et al.
In fact, underground ecosystems are poorly understood in terms of microbiology. Therefore, I always support the study of the cave microbiota and mycobiota, especially if the research goes beyond the usual assessment of species composition. In my opinion, this manuscript is very interesting and well written. I have no typical substantive comments and congratulations on the author of the idea and the manuscript. Nevertheless, I ask the authors to read the manuscript carefully and correct any minor flaws before accepting the manuscript for publication. I have listed some of them below:
Response
Thank you for your appreciation and constructive suggestions in this manuscript. We have acrefully read and corrected all the minor mistake in this revised the manuscript as per your review comments, below.
1) Ln 49: „...sites for CaCO3 precip-...”
“3” – subscript
Response
Thank you for point out this typo mistake. We have corrected it as “ ….. CaCO3…….” in this revised manuscript. ( L 49)
2) Figure 1: Please remove "hrs" from the chart axis.
Response
Thank you for this suggestion. We think “hrs” is incubation period unit. So, we keep as it is. But to make it more scientific we changed the horizontal axis title as “incubation period” in figure 2 of this revised version of manuscript.
3) Figs. 4 and 5: not readable
Response
Thank you for this recommendation. We are very sorry for our inconvenience. We have improved the quality of “Figure 5 and 6”and made them more readable in this new revised manuscript.
Reviewer 2 Report
Dear Authors! I with great pleasure read your article entitled “Assessment of carbon substrate catabolism pattern and functional metabolic pathway for microbiota of limestone caves”. The manuscript presents interesting studies on the assessment of changes in bacterial biodiversity in limestone caves. All relevant scientific preliminaries were provided, the methods were adequately described.
The results are clearly presented, with good conclusions. All tables and graphs are clear, understandable and necessary. The overall quality of the presentation is good. The references are sufficient and necessary. Below I present a few comments regarding figures, statistical analysis and references. In my opinion, these comments can improve the quality of your paper.
Several similar investigations have been carried out recently, dealing with different sampling places, so the novelty of the present work is average. Besides, the interest to the readers seems to be high, because of the well selected place (limestone caves).
I don't feel qualified to review the English used by the authors but I understood well all the text. I haven't found any mistakes.
The article is well written, but I have some suggestions:
- Do you have a photos of sampling places? If yes, please add them in text of paper. For readers it will be interesting to see this sampling places.
- Line 165 – please change “16S rRNA”
- Statistics are not included in Table 1. Please attach some calculations. Line 218
- The quality of the charts is very poor. It's hard to read what's wrong with them. Please correct the charts as recommended by the journal. This is especially true for Figures 4 and 5.
- Do the authors have chemical results of the samples (reaction or other) that could be correlated with the results of Biolog and NGS?
- Please format the text starting from line 279
- Please see some of the publications below to help you discuss your results. Similar studies have been performed in these publications.
Microplot long-term experiment reveals strong soil type influence on bacteria composition and its functional diversity. Applied Soil Ecology, DOI10.1016/j.apsoil.2017.10.033.
Catabolic Fingerprinting and Diversity of Bacteria in Mollic Gleysol Contaminated with Petroleum Substances. Applied Science 2018, 8, 1970; doi:10.3390/app8101970.
Community-level physiological profiles of microorganisms from different types of soil characteristic to Poland – a long-term microplot experiment. Sustainability. 2019, 11, 56., https://doi.org/10.3390/su11010056.
The effect of natural bioremediation on genetic and functional diversity of bacterial microbiome in soils with long – term impacts from petroleum. Front. Microbiol. 9:1923.doi: 10.3389/fmicb.2018.01923.
Wish good luck in your current and future studies.
Author Response
Comments and Suggestions for Authors
Dear Authors! I with great pleasure read your article entitled “Assessment of carbon substrate catabolism pattern and functional metabolic pathway for microbiota of limestone caves”. The manuscript presents interesting studies on the assessment of changes in bacterial biodiversity in limestone caves. All relevant scientific preliminaries were provided, the methods were adequately described.
The results are clearly presented, with good conclusions. All tables and graphs are clear, understandable and necessary. The overall quality of the presentation is good. The references are sufficient and necessary. Below I present a few comments regarding figures, statistical analysis and references. In my opinion, these comments can improve the quality of your paper.
Several similar investigations have been carried out recently, dealing with different sampling places, so the novelty of the present work is average. Besides, the interest to the readers seems to be high, because of the well selected place (limestone caves).
I don't feel qualified to review the English used by the authors but I understood well all the text. I haven't found any mistakes.
The article is well written, but I have some suggestions:
Response
Thank you for your appreciation and constructive suggestions in this manuscript. We have completely revised the manuscript as per your review comments, below.
- Do you have a photos of sampling places? If yes, please add them in text of paper. For readers it will be interesting to see this sampling places.
Response
Thank you for this suggestion. Yes, we have sampling photos and those already added in revised version of this manuscript entitled as “Figure 1. Sampling area and site description”. (L 128)
- Line 165 – please change “16S rRNA”
Response
Thank you for point out this typo mistake. We have changed it as “16S rRNA” in this revised manuscript. (L167)
- Statistics are not included in Table 1. Please attach some calculations. Line 218
Response
Thank you so much for this recommendation. We have included the significant level of student t-Test “P value” in Table 1 of this revised manuscript. Besides mentioned the purpose of using this statistical analysis in material method part. (L 183 – 185)
“Student T-test was used to shown significant difference between outside and inside cave microbiota substrate utilization AWCD value, Shannon index (H'), Simpson index (D) and Shannon evenness score in soil and rock samples”.
“For attach some calculation”, we already revised the original meaning of this sentence as “Additionally, Additionally, the positive control was also showed a certain amount of AWCD value during incubation period, while no AWCD value was found for the negative control.” in this revised manuscript. (L116 to 218)
How to calculate “ACWD” the formula was described in reference study of this manuscript. (L 149)
- The quality of the charts is very poor. It's hard to read what's wrong with them. Please correct the charts as recommended by the journal. This is especially true for Figures 4 and 5.
Response
Thank you for this recommendation. We are very sorry for our inconvenience. We have improved the quality of charts in manuscript’s, especially quality of “Figure 5 and 6” and made them more readable in this new revised manuscript.
- Do the authors have chemical results of the samples (reaction or other) that could be correlated with the results of Biolog and NGS?
Response
Thank you for this suggestion. Unfortunately, this study doesn’t do any chemical analysis, because the study objective was to categorized the microbial community diversity and their metabolic activity in inside and outside cave of a limestone rich area via community level physiological profiling and 16S amplicon based taxonomical and functional metabolic prediction. Although, we have tried to make a correlation between Biolog Ecoplate carbon substrate utilization and relevant predicted functional pathways abundance in all over the samples, the was visualization showed in Figure 6 of this revised manuscript. We hope, this kind of concept is quite novel and never been reported in this field of study.
- Please format the text starting from line 279
Response
Thank you for this comment. We are very sorry for our inconvenience. We have corrected all the formatting issues in this revised manuscript.
- Please see some of the publications below to help you discuss your results. Similar studies have been performed in these publications.
Microplot long-term experiment reveals strong soil type influence on bacteria composition and its functional diversity. Applied Soil Ecology, DOI10.1016/j.apsoil.2017.10.033.
Catabolic Fingerprinting and Diversity of Bacteria in Mollic Gleysol Contaminated with Petroleum Substances. Applied Science 2018, 8, 1970; doi:10.3390/app8101970.
Community-level physiological profiles of microorganisms from different types of soil characteristic to Poland – a long-term microplot experiment. Sustainability. 2019, 11, 56., https://doi.org/10.3390/su11010056.
The effect of natural bioremediation on genetic and functional diversity of bacterial microbiome in soils with long – term impacts from petroleum. Front. Microbiol. 9:1923.doi: 10.3389/fmicb.2018.01923.
Wish good luck in your current and future studies.
Response
Thank you for recommendation to go through these publications and appreciate this study. We have read all the articles and it help us to make a better idea on how to describe the microbial community level physiological profiling in respective environmental samples. Among of them, two articles have been cited in this revised version of manuscript, reference no 23 and 30. (L 83, 154)
- “GrzÄ…dziel, J.; Furtak, K.; Gałązka, A. Community-level physiological profiles of microorganisms from different types of soil that are characteristic to Poland—a long-term microplot experiment. Sustainability 2019, 11, 56.”
- “GrzÄ…dziel, J.; Gałązka, A. Microplot long-term experiment reveals strong soil type influence on bacteria composition and its functional diversity. Applied Soil Ecology 2018, 124, 117-123.”
Round 2
Reviewer 2 Report
The authors responded to all my comments and revised the manuscript significantly. Accepts papaer in its current form.